# Phase I trial of the TNF-α inhibitor certolizumab plus chemotherapy in stage IV lung adenocarcinomas

Paul K. Paik [1,2] ✉, Jia Luo [1], Ni Ai[3], Rachel Kim[1], Linda Ahn[1], Anup Biswas[4], Courtney Coker [4], Wanchao Ma[4], Phillip Wong [5], Darren J. Buonocore[6], W. Victoria Lai[1,2], Jamie E. Chaft[1,2], Swarnali Acharyya[4,7,8], Joan Massagué[9] & Mark G. Kris[1,2]

We previously identified a chemotherapy-induced paracrine inflammatory loop that paradoxically mitigates the anti-tumor effect of chemotherapy and triggers metastatic propagation in breast and lung cancer models. Therefore, we sought to further validate and translate these findings into patient care by coupling the anti-TNF-α drug certolizumab pegol with standard cisplatin doublet chemotherapy. Here we first validate the anti-metastatic effect of certolizumab in a liver-metastatic Lewis Lung Carcinoma model. We then evaluate the safety, efficacy, and pharmacodynamic effects of certolizumab with cisplatin and pemetrexed in an open label Phase 1 clinical trial (NCT02120807) of eighteen adult patients with stage IV lung adenocarcinomas. The primary outcome is maximum tolerated dose. Secondary outcomes are response rate and progression-free survival (PFS); pharmacodynamic changes in blood and tumor are evaluated as a correlative outcome. There were nine partial responses among 16 patients evaluable (56%, 95% CI 30 to 80%). The median duration of response was 9.0 months (range 5.9 to 42.6 months) and median PFS was 7.1 months (95% CI 6.3 to NR). The standard 400 mg dose of certolizumab, added to cisplatin and pemetrexed, is well-tolerated and, as a correlative endpoint, demonstrates potent pharmacodynamic inhibition of peripheral cytokines associated with the paracrine inflammatory loop.

The discoveries of therapies targeting oncogenic driver mutations and immune checkpoint inhibitors (ICIs) have revolutionized the care of patients with metastatic lung cancers. Yet despite high initial response rates and improvements in survival for many patients, metastatic lung cancer remains incurable. Cytotoxic chemotherapy remains the standard for patients whose cancers lack an actionable oncogenic driver or develop acquired resistance to targeted agents or immunotherapies[1]. ICIs alone are not efficacious for the majority of patients[2] and demonstrate even less benefit in individuals whose tumors harbor certain oncogenic drivers[3,4]. Cytotoxic chemotherapy continues to be employed routinely, now with ICIs and particularly in symptomatic patients, although the biologic rationale and implications of this strategy are not understood. These data highlight the critical need to uncover the mechanisms underlying resistance to chemotherapy and metastasis propagation and develop strategies to thwart them.

Our prior work identified a paracrine inflammatory loop in breast cancers that disrupts the microenvironment of both the tumor and surrounding myeloid cells and involves TNF-α, CXCL1/2, and S100A8[5]. Briefly, treatment with chemotherapy induces cellular stress in the stromal endothelial cells that generates TNF-α. Endothelial cell-

**Fig. 1 | Anti-TNF-α antibodies enhance the anti-metastatic effect of cisplatin.**
**a** Schematic of intra-cardiac (i.c.) injections and development of metastasis in the murine LLC liver metastasis lung cancer model. **b** A total of $1 \times 10^5$ murine LLC-liver met cells were injected into the arterial circulation via i.c. injection in 8-to-9-week-old syngeneic C57BL/6 male mice, which gradually developed metastasis to the liver, lymph nodes and chest cavity, as confirmed by bioluminescence imaging on day 16. **c** Mean normalized photon flux bioluminescence levels following treatment as indicated for each group (biologically independent mice per group - $n = 6$: cisplatin + anti-TNF, $n = 7$: isotype, cisplatin, anti-TNF). Source data are provided as a Source Data file. Values are mean ± standard error of unpaired, two-tailed Student's $t$-test. *$p = 0.0012$, **$p = 0.007$, ns not significant.

## Table 1 | Baseline clinicopathologic characteristics

|  | Patients $n = 18$ |
| --- | --- |
| Age, median [range] - years | 63 [42–71] |
| Gender |  |
| Female | 11 (63%) |
| Histology |  |
| Adenocarcinoma | 18 (100%) |
| Smoking history |  |
| Never | 1 (6%) |
| Current/former | 17 (94%) |
| Median pack years [range] | 52 [0–456] |
| Karnofsky performance status, median [range] | 80 [70–90] |
| Driver mutation status |  |
| Non-canonical *EGFR* alteration |  |
| *EGFR* S720F + L861Q | 1/14 (7%) |
| *EGFR* kinase domain duplication | 1/14 (7%) |
| *KRAS* mutation | 5/14 (36%) |
| *ALK* rearrangement | 1/14 (7%) |

generated TNF-α then triggers CXCL1/2 expression by cancer cells, which subsequently recruits CD11b+Gr1+ myeloid-derived suppressor cells that express CXCR2 (the receptor for CXCL1/2). These myeloid cells produce S100A8, which enhances cancer cell survival and triggers metastatic seeding (Supplementary Fig. 1). As a result, tumors become insensitive to chemotherapy and highly metastatic. Dr. Acharyya demonstrated that blocking the paracrine inflammatory loop at any level is sufficient to abrogate chemoresistance and metastatic propagation[5]. Components of this paracrine inflammatory loop are also activated in lung cancer[6–8]. TNF-α is increased in the serum of patients with lung cancers and animal models of lung adenocarcinomas[9,10]. Given the phenotypic overlap, we hypothesized that a similar strategy may improve the efficacy of chemotherapy and interrupt the biologic mechanisms that underpin metastasis formation in lung cancer.

Multiple anti-TNF-α agents, including certolizumab, infliximab, and adalimumab, have become the mainstay of treatment for rheumatoid arthritis and inflammatory bowel disease. Anti-TNF-α agents have posed no additional safety issues when given to persons with lung cancers and in patients with other solid tumors receiving chemotherapy[11]. Thus, TNF-α inhibitors may enhance chemotherapy effectiveness and disrupt the metastatic process in lung cancer. Here, certolizumab is an ideal candidate drug, as it is the only TNF-α antagonist utilizing the Fab fragment of a humanized TNF-α antibody which lacks the Fc region[12]. This avoids both complement-dependent cytotoxicity and antibody dependent cell mediated cytotoxicity as mechanistic confounders present with all other anti-TNF agents. We therefore conducted additional preclinical studies and a phase I clinical trial to test the efficacy of certolizumab with cisplatin chemotherapy in patients with lung cancers. In this work, we validate the anti-metastatic effect of certolizumab in a preclinical model of lung cancer and demonstrate tolerability and preliminary efficacy results in a phase I clinical trial.

## Results

### Combination of anti-TNF-α and cisplatin therapy reduces metastasis and increases anti-tumor efficacy

Based on previous studies in breast cancer[5], we sought to test if the combination of chemotherapy and TNF-α inhibition can also be effective in reducing lung cancer metastasis. To this end, we utilized an experimental metastasis model of lung cancer derived from the murine LLC cell line, LLC-liver met. Murine LLC-liver met cells were injected into the arterial circulation of C57BL/6 mice via intra-cardiac (i.c.) injection (Fig. 1a), which gradually developed metastasis to the liver, lymph nodes, and chest cavity, as confirmed by bioluminescence imaging (Fig. 1b). Treatment with cisplatin, and TNF-α antibody to a lesser extent, reduced metastasis over isotype-control treated mice ($p = 0.007$; Fig. 1c), with the combination of cisplatin plus anti-TNF-α antibody further lowering metastatic burden ($p = 0.0012$), providing a rationale for combining TNF-α inhibitors with cisplatin chemotherapy in clinical studies. An orthotopic lung metastatic CMT-167 murine lung cancer model was also tested in syngeneic C57BL/6 mice via lung injection using the same treatment conditions (Supplementary Fig. 2), with the addition of anti-TNF-α antibody to cisplatin leading to significant reduction in lung tumor burden.

### Patient characteristics

Eighteen patients with stage IV lung adenocarcinomas were treated, with clinicopathologic characteristics summarized in Table 1. The median age of the patients was 63, 63% percent were female, and 94%

were current or former smokers, with a median smoking history of 52 pack-years. Sufficient tumor material was available from fourteen patients treated with certolizumab (14/18; 78%) for *EGFR, KRAS, BRAF, ERBB2,* and *ALK* testing and from ten patients (10/18; 56%) for NGS (Supplementary Table 1). With regard to known oncogenic alterations, five had a *KRAS* mutation (5/14; 36%), two had non-canonical *EGFR* mutations (2/14; 14%), and one had an *ALK* rearrangement (1/14; 7%). Genotype data was available for 9/10 patients in the control cohort, with four harboring *KRAS* mutations (4/9, 44%) (Supplementary Table 1).

## Dose delivery

We planned six cycles of cisplatin and pemetrexed and 6 doses of certolizumab. The median number and range of doses delivered on study for each drug during the triplet combination administration were as follows: certolizumab = 6 (range 3 to 6), cisplatin = 6 (range 3 to 6), and pemetrexed = 6 (range 3 to 6). Three patients were treated

with certolizumab 200 mg. Fifteen patients were treated with certolizumab 400 mg; three as part of dose escalation, and ten as part of the expansion cohort. No dose reductions or dose holds were required for any of the therapeutic agents.

## Treatment-related side-effects

All eighteen patients were evaluable for toxicity. The most common any-grade related adverse events (AEs) were nausea (12/18; 67%) constipation (11/18; 61%), anemia (11/18; 61%), and leukopenia (9/18; 50%) (Table 2). Grade ≥3 related AEs occurred in 44% (8/18) of patients and were mostly hematologic: neutropenia (4/18; 22%); leukopenia (1/18; 6%); and thrombocytopenia (1/18; 6%). No febrile neutropenia was seen. No DLTs were observed during dose escalation or dose expansion. There were three unrelated serious AEs (grade 2 lung infection, grade 2 skin infection, and grade 5 cardiac arrest). Safety was similar across certolizumab doses.

## Efficacy

Sixteen patients were evaluable for response. Nine partial responses were observed (9/16; 56%, 95% CI 30 to 80%) (Fig. 2a). The median duration of response was 9.0 months (range 5.9 to 42.6 months). The median PFS was 7.1 months (95% CI 6.3 to NR) (Fig. 2b, Supplementary Fig. 1A). The median OS was 22.6 months (95% CI 12.7 to NR) (Supplementary Fig. 1B).

## Cytokine assessments

Fifteen patients had baseline and at least one on-treatment blood draw for cytokine analysis. Serial peripheral cytokine assessment by ELISA demonstrated significant, marked, and early suppression of TNF-α in patients treated with certolizumab and chemotherapy vs. chemotherapy alone (5.3-fold decrease vs. 1.02-fold increase, $p < 0.001$) (Fig. 3a, b). S100A8 levels decreased over time in the treatment group and increased over time in the control group: the nadir, mean, and AUC for S100A8 were significantly different between patients treated with certolizumab and chemotherapy vs. the control group ($p = 0.01$, $p = 0.02$, $p = 0.02$, respectively) (Fig. 3c).

Nadir TNF-α was significantly associated with PFS (for every log decrease in TNF-α, HR = 0.4, 95% CI 0.17–0.96, $p = 0.04$) and exhibited

**Table 2 | Treatment-related adverse events greater than 20%**

|                    | Grade 1  | Grade 2  | Grade 3  | Grade 4 | All Grades |
|--------------------|----------|----------|----------|---------|------------|
| Nausea             | 9 (50%)  | 2 (11%)  | 1 (6%)   | –       | 12 (67%)   |
| Constipation       | 9 (50%)  | 2 (11%)  | –        | –       | 11 (61%)   |
| Anemia             | 1 (6%)   | 9 (50%)  | 1 (6%)   | –       | 11 (61%)   |
| Leukopenia         | 1 (6%)   | 7 (39%)  | 1 (6%)   | –       | 9 (50%)    |
| Neutropenia        | –        | 4 (22%)  | 3 (17%)  | 1 (6%)  | 8 (44%)    |
| Thrombocytopenia   | 5 (28%)  | 2 (11%)  | 1 (6%)   | –       | 8 (44%)    |
| Fatigue            | 5 (28%)  | 2 (11%)  | –        | –       | 7 (39%)    |
| Neuropathy         | 6 (33%)  | –        | –        | –       | 6 (33%)    |
| Dyspnea            | 4 (22%)  | 1 (6%)   | –        | –       | 5 (28%)    |
| Hyponatremia       | 4 (22%)  | –        | 1 (6%)   | –       | 5 (28%)    |
| Vomiting           | 3 (17%)  | 1 (6%)   | –        | –       | 4 (22%)    |
| Diarrhea           | 3 (17%)  | 1 (6%)   | –        | –       | 4 (22%)    |
| Anorexia           | 2 (11%)  | 2 (11%)  | –        | –       | 4 (22%)    |
| Creatinine increase| –        | 4 (22%)  | –        | –       | 4 (22%)    |
| Hypomagnesemia     | 3 (17%)  | 1 (6%)   | –        | –       | 4 (22%)    |

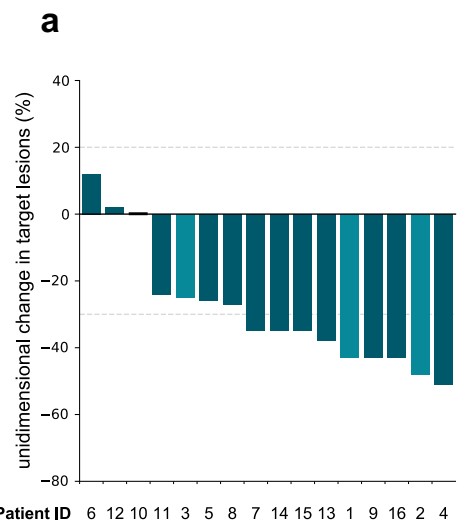

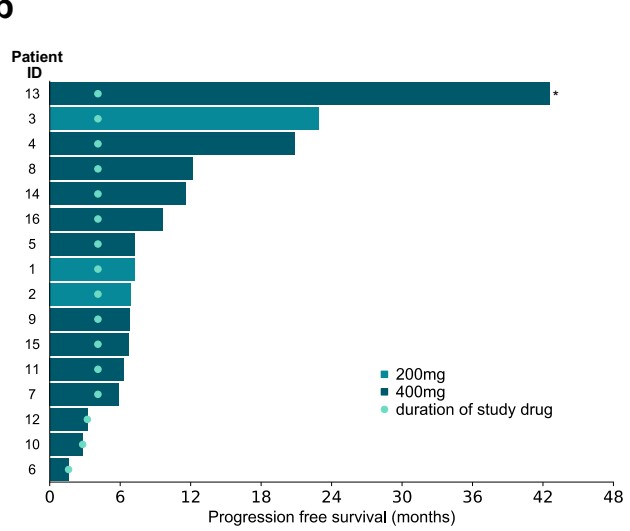

*data censored at last date of follow-up

**Fig. 2 | Radiographic responses and progression-free survival for patients with advanced lung adenocarcinomas. a** Best unidimensional response per RECISTv1.1. **b** Swimmer plot depicting progression-free survival (PFS) for patients treated with cisplatin, pemetrexed, and certolizumab (median PFS 7.1 months, 95%

CI 6.30 to NR). Data reflects progression in all patients (*n* = 16) save for patient 13, who was censored at the date of last follow-up. Source data are provided as a Source Data file.

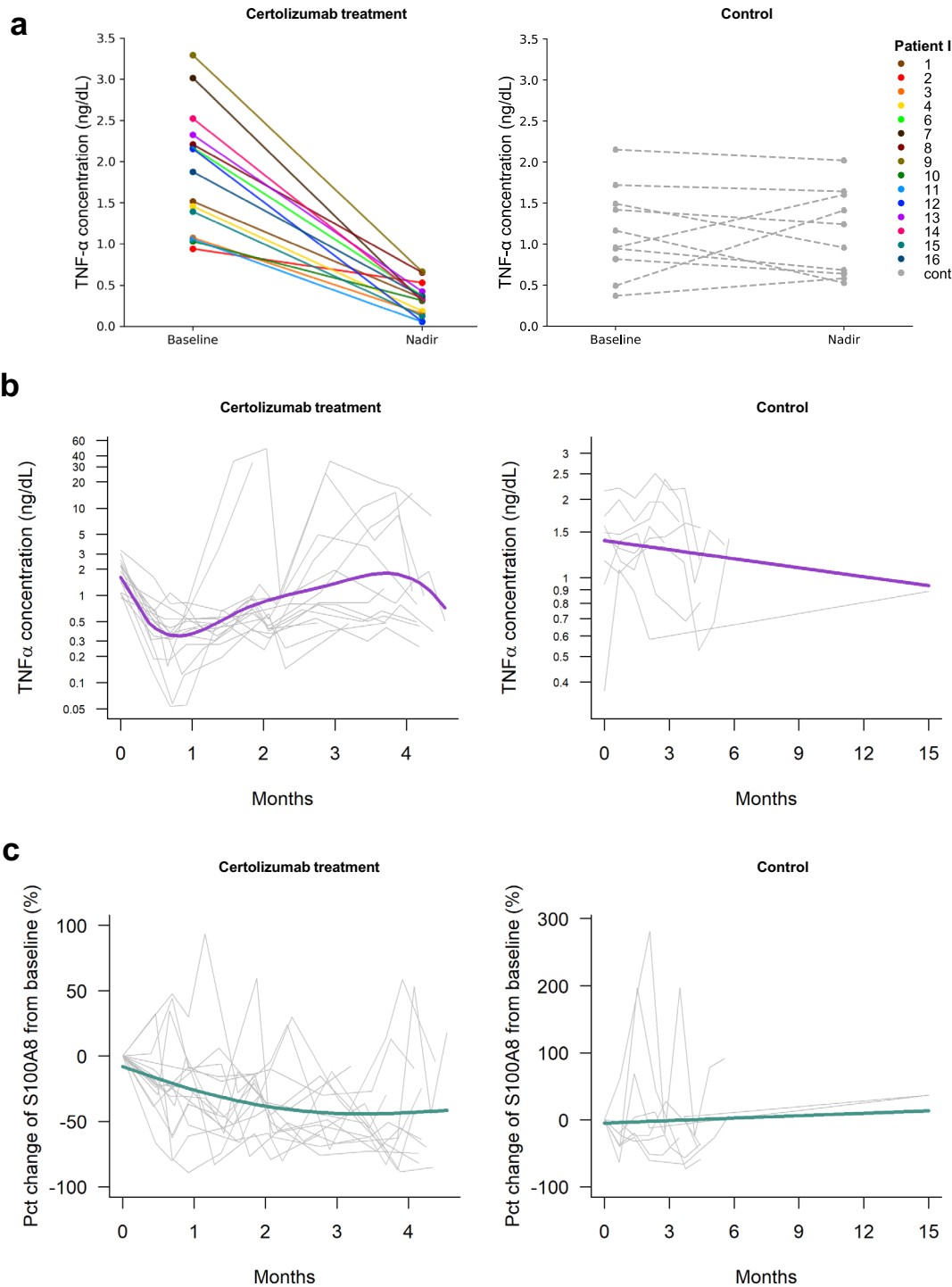

**Fig. 3 | Longitudinal TNF-α and S100A8 concentrations during therapy. a** Trend plots depicting changes in TNF-α concentration from baseline to nadir during the course of therapy for patients (*n* = 15) treated with cisplatin, pemetrexed, and certolizumab (left panel) and contemporaneous control patients (*n* = 10) treated with platinum plus pemetrexed chemotherapy (right panel). **b** Plots showing individual changes in log TNF-α concentration. **c** Percent (Pct) change in S100A8 concentration in contemporaneous control patients treated with patients treated with cisplatin + pemetrexed + certolizumab (right panels). Source data are provided as a Source Data file.

a trend toward an association with OS (for every log decrease in TNF-α, HR = 0.42, 95% CI 0.17–1.04, *p* = 0.06). TNF-α changes showed no association with PFS or OS in the control group. Nadir S100A8 was also significantly associated with PFS and OS, with every percent increase in S100A8 relative to baseline increasing the HR for PFS by 1.06 (95% CI 1.02–1.11, *p* = 0.008) and HR for death by 1.08 (95% CI 1.02–1.14, *p* = 0.006). Time-dependent modeling was also performed to explore associations between TNF-α and S100A8 and PFS and OS, which showed trends towards worse PFS with increasing TNF-α and worse OS with increasing TNF-α and S100A8 (Supplementary Table 2).

**Immune cell infiltrate**

Eleven patients underwent either pre-treatment or post-treatment protocol-related biopsies to assess immune cell infiltrates. Of these eleven, four patients had sufficient tumor material in pre-treatment and post-treatment samples for an analysis of immune cell infiltration

by immunohistochemistry. The aggregate results per patient are shown in Supplementary Table 3. One patient, patient 3, was treated during the dose-escalation phase at the lower 200 mg certolizumab dose. All other patients were treated at the 400 mg certolizumab dose level. In general, there were no substantial differences in immune cell infiltration in pre- vs. post-treatment samples. The one exception was patient 3, whose tumor showed a marked increase in immune cell infiltration, particularly intratumorally, post-treatment. This included intratumoral increases of 1371% in FOXP3 positive cells (17 to 233 cells/HPF (high power field) and 531% in CD11c positive cells (16 to 85 cells/HPF). IHC photomicrographs are shown in Supplementary Fig. 2.

## Discussion

Our phase I study examining the addition of certolizumab, an anti-TNF-α monoclonal antibody, to cisplatin and pemetrexed chemotherapy in patients with untreated stage IV lung adenocarcinoma demonstrated that this intervention is feasible, well-tolerated, and has a promising efficacy signal. Our study is based on the strong, specific pre-clinical rationale that chemotherapy, while beneficial, also instigates a paracrine inflammatory loop involving the tumor and stroma, whereby cytotoxic stress induces endothelial cell secretion of TNF-α, which promotes cancer-cell secretion of CXCL1/2 and the recruitment of CXCR2-expression myeloid-derived suppressor cells (MDSCs). These MDSCs engender cancer cell chemoresistance and metastatic propagation through S100A8. We note that the overarching objective of our study was to efficiently demonstrate that this paracrine inflammatory loop, which we had previously characterized in exacting mechanistic detail, could be therapeutically targeted using a clinically available drug in (1) lung cancer models in order to reduce metastatic burden and (2) patients with lung cancer in order to improve clinical outcomes.

TNF-α production is the most proximal factor in the chemotherapy-triggered inflammatory loop and, given the self-amplifying effects of the loop, is an attractive therapeutic target that alone is well-tolerated, with a number of FDA-approved blocking monoclonal antibodies available in other, largely rheumatologic, indications. We found evidence of a combinatorial anti-metastatic effect in two immunocompetent lung cancer metastatic models. Single agent TNF-α inhibition also yielded an anti-metastatic effect to varying degrees across models and animals, suggesting that TNF-α may also have a more proximal effect in the metastatic colonization process as previously suggested[13]. No previously unreported or additive safety signals were seen in our study with the addition of certolizumab to cisplatin and pemetrexed. Importantly, no DLTs or infectious complications were encountered. Indeed, the regimen was well-tolerated, with only a 11% rate of treatment-related non-hematologic AEs reported and no dose reduction or dose holds required.

While conventional theory, largely unfounded and contradicted by our specific pre-clinical data, might raise the possibility that the immunosuppressive effects of TNF-α inhibition could impede an anti-tumor response, our study does not bear this out. The ORR of 56%, median PFS of 7.1 months, and median OS of 22.6 months compare favorably to the robust historical data for platinum plus pemetrexed, which is associated with an ORR of 30.6%, median PFS of 4.8–5.3 months, and median OS of 10.3–12.6 months in an immune checkpoint inhibitor naïve population[1]. Our data also compare favorably with a more contemporaneous control population treated with carboplatin plus pemetrexed and maintenance pemetrexed in KEYNOTE 189, which recorded an ORR of 18.9%, median PFS 4.9 months, and median OS 11.3 months[14]. Our study population was similar both in clinical characteristics and genotype distribution to other study populations, with a median age of 63, most having a smoking history, and *KRAS* mutations in 36%.

The correlative data indicate that the hypothesized effect on the paracrine inflammatory loop was seen, both in terms of an immediate

and marked down-regulation of peripheral TNF-α and distal inhibition of S100A8. Both of these markers were also associated with PFS (higher concentrations portended worse PFS), supporting, though not cementing, a potential casual effect from certolizumab as suggested by our hypothesis. While angiogenesis inhibition was utilized as part of the regimen for some patients in the control cohort, this was a factor difficult to control for, as at the time bevacizumab was part of the standard first-line regimen at MSK for patients with lung adenocarcinoma (and so withholding it could not be mandated), though we have no a priori expectations that angiogenesis inhibition would directly affect TNF-α levels. Limited IHC for general immune markers in matched pre- and post-treatment biopsies also suggested that certolizumab might have had an effect on the tumor microenvironment. One patient treated at the sub-therapeutic dose of 200 mg was the only patient to have had matched biopsies showing a substantial post-treatment influx of immune cells, including FOXP3-expressing T-cells and CD11c + cells. These data, while aligning with our hypothesis, will need to be confirmed.

Although this report did not study the role of TNF-α inhibition with immune checkpoint-based therapy, a brief discussion is warranted given the current treatment landscape for lung cancers. Investigators are exploring TNF-α suppression to mitigate immune therapy related side effects and to boost efficacy based in part on inhibition of MDSC trafficking. Retrospective data in melanoma, have shown that anti-TNF-α agents do not compromise the efficacy of immune checkpoint inhibitors when administered to combat immune-mediated colitis caused by ICIs[15,16]. Furthermore, pre-clinical data demonstrate that TNF-α inhibition improves the tolerability of ICIs while increasing their efficacy[17,18]. This approach has generated promising data in a trial of certolizumab added to ipilimumab and nivolumab in patients with metastatic melanoma, with a reported objective response rate of 88% NCT03293784[19]. Our study adds to this literature and lays the groundwork for incorporating TNF-α inhibition with chemotherapy. This is important, as most patients with lung cancers will receive treatment with platinum doublet chemotherapy added to an immune checkpoint inhibitor. Our study shows that the addition of TNF-α inhibition to chemotherapy adds no additional toxicities and can serve as the foundation for future studies adding certolizumab to chemotherapy plus immune checkpoint inhibition.

This study was the necessary step towards translating our pre-clinical findings to patients. Further studies are planned to validate the observation of better efficacy and tolerance of certolizumab plus chemotherapy, assess the impact of the combination on retarding the development of metastatic disease, and better evaluate its impact on the immune microenvironment. These objectives will next be pursued in our phase 2 trial of neoadjuvant certolizumab + platinum doublet chemotherapy + nivolumab in patients with resectable stage II-IIIA lung cancers (NCT04991025).

## Methods

### In vivo lung cancer modeling using murine studies

All mouse protocols were approved by The Columbia University (New York, NY) Institutional Animal Care and Use Committee (IACUC) and conducted according to the guidelines from Columbia University Medical Center's (CUMC) Institute of Comparative Medicine (ICM). The maximal tumour size/burden permitted by the Columbia IACUC is 2 cm in diameter; no mice met this criterion. Following IACUC guidelines from Columbia University, weight loss of more than 20%, body conditioning score (BCS) of 2 or less, or mice exhibiting signs of hunched posture, impaired locomotion or respiratory distress are criteria followed for prompt euthanasia. Mice with a BCS of less than 3 are monitored by the investigative staff at least daily and provided with additional supportive care or as directed by an ICM veterinarian. Euthanasia with carbon dioxide inhalation with secondary method of decapitation/cervical dislocation was followed. Mice were housed

within CUMC's specific pathogen-free barrier facility with regulated environment and fed Labdiet 5053 (standard diet). Male, 8-to-9-week-old C57BL/6 mice were purchased from Jackson Laboratory (Bar Harbor, Maine). A liver metastatic subline of Lewis lung carcinoma line (LLC-liver met) was generated in the Acharyya laboratory by the in vivo selection method[20]. C57BL/6 mice were injected with $1 \times 10^5$ luciferase-labeled LLC-liver met cancer cells by intra-cardiac injection into the arterial circulation. Metastasis was monitored weekly by bioluminescence imaging. Mice were randomized into four groups with 6–8 mice per group: (1) isotype control antibody, (2) cisplatin chemotherapy, (3) anti-TNF-α antibody, and (4) cisplatin + anti-TNF-α antibody combination. Isotype (cat #BP0088) and anti-TNF-α (cat #BP0058) antibodies were purchased from Bio X Cell (Lebanon, New Hampshire), diluted in phosphate buffered saline, and injected intraperitoneally at 200μg antibody per mouse. Antibody treatment started four days post-tumor cell injection and was continued thereafter three times a week until the conclusion of the study. Cisplatin chemotherapy (Sigma, St. Louis, Missouri) was injected intraperitoneally at 3 mg/kg at day seven post-tumor cell injection followed by 5 mg/kg at day 14. Final images of bioluminescence were taken at day 16 post-tumor cell injection and analyzed using Living Image software (Perkin Elmer, Waltham, Massachusetts).

For orthotopic lung injections, mice were anesthetized with isoflurane, which was administered with a precision vaporizer[21]. Depth of anesthesia was monitored at least every 15 min throughout the procedure by observing that there was no change in respiratory rate associated with surgical manipulation and/or toe pinch. Mice were placed on a snuggle safe pad with a barrier between mouse and pad. Following confirmation that a suitable anesthetic plane (no response to stimulation) had been attained, sterile eye lubricant was applied to both eyes to prevent corneal drying. The area around the injection site was shaved, and sterility prepped with Betadine and 70% alcohol consisting of three alternate swabs. A 5 mm incision was made on the side of the left thoracic cavity. One-ml tuberculin syringes were used to inject the cell inoculum into the left lateral thorax, at the lateral dorsal axillary line, approximately 1.5 cm above the lower rib line just below the inferior border of the scapula once the lungs were visualized through the ribs. The needle was quickly advanced approximately 6 mm into the thorax and injected into the lungs while they were extended. A total of 30 μl of luciferase-labeled CMT-167 cancer cells (cat #10032302-1VL, Sigma–Aldrich, Inc. St. Louis, MO) in Hanks buffered saline ($1 \times 10^5$ cells) were injected. The needle was quickly removed after the injection. The muscles were sutured together using Vicryl sutures. Wound clips were used to close the skin. The procedure took 10–15 min per mouse. The mice were imaged by bioluminescence imaging. Mice were randomized into four groups: (1) isotype control antibody, (2) cisplatin chemotherapy, (3) anti-TNF-α antibody, and (4) cisplatin + anti-TNF-α antibody combination. Isotype (cat #BP0088) and anti-TNF-α (cat #BP0058) antibodies were purchased from Bio X Cell (Lebanon, New Hampshire), diluted in phosphate-buffered saline, and injected intraperitoneally at 200 microgram antibody per mouse. Antibody treatment started four days post-tumor cell injection and was continued thereafter three times a week until the conclusion of the study. Cisplatin chemotherapy (Sigma, St. Louis, Missouri) was injected intraperitoneally at 3 mg/kg on day seven post-tumor cell injection. Final images of bioluminescence were taken on day 14 post-tumor cell injection and analyzed using Living Image software (Perkin Elmer, Waltham, Massachusetts). Normalized photon flux was calculated by dividing photon flux from Day 14/Day 0 (day of injection).

### Study design and participants
This single-center, open-label, single-arm, phase I trial (NCT02120807) using certolizumab in combination with chemotherapy in patients with stage IV lung adenocarcinomas was reviewed and approved by the Institutional Review Board at Memorial Sloan Kettering (MSK, New York, NY) and conducted in accordance with the Declaration of Helsinki and Good Clinical Practice guidelines. All patients provided written informed consent prior to participation.The first and last patient enrollments occurred on April 22, 2014 and January 27 2016, respectively. Follow-up for overall survival (OS) is ongoing as of February 2021. Eligibility criteria included KPS ≥ 70%, no prior systemic therapies, pathologically confirmed metastatic or recurrent lung adenocarcinomas, absolute neutrophil count $\geq 1.5 \times 10^9/L$, hemoglobin ≥ 8 g/dL, platelets $\geq 100 \times 10^9/dL$, total bilirubin ≤1.5 x upper limit of normal (ULN), aspartate transaminase/alanine aminotransferase ≤2.5 x ULN, serum creatinine ≤1.5 x ULN. Exclusion criteria included autoimmune disorder, prior use of TNF-α inhibitors, systemic corticosteroids, known immunodeficiency syndrome, or another active cancer.

### Procedures
Patients received cisplatin 75 mg/m² and pemetrexed 500 mg/m² intravenously every three weeks for six cycles. Starting on the same day as chemotherapy, all patients also received certolizumab subcutaneously every two weeks for one month, and then monthly thereafter. Patients were enrolled in a 3 + 3 dose escalation scheme using two dose levels of certolizumab – 200 mg − 50% of the FDA-approved dose; and 400 mg in combination with a de-escalation scheme for chemotherapy if necessary, followed by a dose expansion cohort at the maximum tolerated dose (MTD) of the combination. Additional cycles of maintenance pemetrexed-based therapy were provided per the patient's treating physician per standard of care.

Patients were monitored for progression of disease using RECIST v1.1 with CT scans every six weeks. Patients who completed all six cycles of therapy completed the study requirements. Pemetrexed 500 mg/m² based therapy was continued every three weeks until disease progression at the discretion of the treating physician. Toxicities were graded using the National Cancer Institute Common Toxicity Criteria for Adverse Events (CTCAE) version 4.0. Biopsies were collected at the time of treatment discontinuation to provide tissue for pharmacodynamic/biomarker studies.

A separate control cohort of ten patients undergoing standard of care platinum-based chemotherapy plus pemetrexed with or without bevacizumab were consented to blood draws before each chemotherapy treatment for cytokine analysis. This cohort served as a reference for the experimental arm (Supplementary Table 4). When material was available, archived tumors underwent next-generation sequencing (NGS) by MSK-Integrated Mutation Profiling of Actionable Cancer Targets (MSK-IMPACT)[22].

As part of standard of care, patient tumor samples were tested for oncogenic drivers using a combination of immunohistochemistry (ALK), PCR (EGFR mutations), and MSK IMPACT[22].

### Cytokine response assays
Blood was collected for cytokine analysis by ELISA at baseline and prior to each chemotherapy infusion (day 1 of each cycle) or each certolizumab administration. V-PLEX validated Human Proinflammatory Panel 10-plex kits were purchased from Meso Scale Diagnostics to measure TNF-α (MSD, Kenilworth, New Jersey, Cat #K15049D-1). All reagents were provided with the kit. The standards were reconstituted in the assay diluent provided. Frozen plasma samples were thawed, clarified by brief centrifugation to remove any particulate materials, and diluted two-fold in assay diluent. Diluted samples, controls, and standards were added at 50 μl per well into the multiplex MSD assay plates pre-coated with 10 capture antibodies per well against the cytokines of interest. The plate was sealed and incubated for 2 h at room temperature on an orbital shaker (600 rpm). At the end of the incubation, the wells were washed three times using 150 μl wash buffer (PBS + 0.05% Tween 20). Detection antibodies conjugated to electrochemiluminescent labels were added at 25 μl per well, and the plate sealed and incubated for 2 h

at room temperature on an orbital shaker (600 rpm). At the end of the incubation, the plate was washed three times as before. Then, 150 μl of the MSD Read Buffer was added to each well, and the plates were read on the MSD Meso QuickPlex SQ 120 imager. The raw data was measured as light intensity detected by instrument photodetectors upon application of electricity to the plate electrodes. Data were analyzed using the MSD Discovery Workbench software. A 4-parameter logistic fit calibration curve was generated for each analyte using the standards to calculate the concentration of each sample.

Individual commercial ELISA kits were purchased for S100A8 (Hycult Biotech, Plymouth Meeting, Pennsylvania, Cat #HK379). All standards and reagents were provided with each kit, and the assays were performed according to manufacturer's instructions. Plates were read using a Molecular Devices SpectraMax plate reader, and the data were analyzed using SoftMax Pro v.6 software. A 4-parameter logistic fit calibration curve was generated for each analyte using the standards to calculate the concentration of each sample.

### Immunohistochemistry

Immunohistochemical analysis was performed on archived formalin fixed paraffin embedded (FFPE) patient tumor biopsies (pre-treatment and end of treatment, when available) to evaluate intratumoral and extratumoral immune cell infiltration using nine antibodies. The following antibodies were used: CD45 (clone X16/99; pre-diluted; Leica Biosystems, Wetzlar, Germany, cat #PA0042), CD3 (clone LN10; dilution 1:250; Leica Biosystems, cat #NCL-L-CD3-565), CD4 (clone SP35; dilution 1:12.5; Cell Marque, Rocklin, California, cat #104R-16), CD8 (clone SP57, pre-dilute, Ventana medical systems, Oro Valley, Arizona, cat #790-4460), CD56 (clone MRQ42, pre-dilute; Cell Marque, cat #760-4596), CD11c (clone EP1347Y; 1:200; Abcam, Cambridge, United Kingdom, cat #ab52632); CD25 (clone 4C9; pre-diluted; Leica Biosystems, cat #PA0305); CD14 (clone EPR3653; pre-dilute; Cell Marque, cat #760-4523), and FOXp3 (clone 236 A/E7; 1:500; Abcam, Cambridge, United Kingdom, cat #AB20034). For each biopsy specimen, the number immune cells were assessed by averaging the number of positive cells present within tumor over ten high power fields (400X). Extratumoral immune cells in the adjacent tissue were assessed by the same process. PD-L1 (clone E1L3N; 1:100; Cell Signaling Technology, Danvers, Massachusetts; #13684) expression was quantified, and the percentage of tumor cells showing membranous staining was determined.

### Statistical analysis

Statistical analyses for murine studies were performed by unpaired, two-tailed Student's $t$-test.

The primary endpoint of the clinical trial was to determine the maximum tolerated dose (MTD) of certolizumab when given in combination with cisplatin and pemetrexed. The MTD was defined as the dose for which ≤1/6 patients had a dose limiting toxicity (DLT), which included grade 4 neutropenia, febrile neutropenia, and grade ≥3 infection. Secondary objectives included objective response rate (ORR) by RECIST v1.1, progression-free survival (PFS), and OS. Medians of PFS and OS were estimated by the Kaplan–Meier method. Correlative objectives included evaluation of the on-target effect of certolizumab on the TNF-α / CXCL1-2 / S100A8/9 axis through serial quantification of serum TNF-α and S100A8 concentrations. Pre- and post-treatment biopsy specimens were analyzed for changes in intratumoral immune cell infiltrates and compared qualitatively. Nadir, mean, and area under the curve (AUC) of individual patients' cytokine trajectory were compared between treated and control patients using the Wilcoxon rank sum test. AUC was calculated using the trapezoidal rule and then standardized by dividing it by the length of trajectory. These quantities were also correlated with PFS and OS using a Cox proportional hazards model in both time-independent and time-dependent models. Smooth curves were generated using cubic splines with 4 knots. All statistical analyses

were conducted in R v4.0.3 (R foundation for Statistical Computing, Vienna, Austria) or GraphPad Prism 8 (San Diego, California). Two-sided $P$ values < 0.05 were considered statistically significant.

### Reporting summary

Further information on research design is available in the Nature Research Reporting Summary linked to this article.

## Data availability

The datasets generated during and/or analyzed during the current study are available in the Article, Supplementary Information and Source Data file. Clinical data are deidentified. All shareable, individual de-identified clinical data, including radiographic response data, time-dependent endpoint data, blood cytokine data, and IHC tumor data are provided in the Source Data file. Study protocol can be provided on request. Source Data are provided with this paper.

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

## Acknowledgements

This study was supported in part by the MSK Lung P01 CA-129243 and P30 CA-008748; the National Institutes of Health [T32-CA009207 to J.L., K30-UL1TR00457 to J.L., R00-CA172697 to S.A.]; the Conquer Cancer Foundation [Young Investigator Award to J.L.]; a Society of MSKCC grant; a DoD Lung Cancer Metastasis grant [W81XWH-17-1-0441 to S.A.]; a Herbert Irving Scholar Award to S.A.; and the Ludwig Center for Cancer Immunotherapy for support of the Immune Monitoring Facility at MSK. The funders had no role in study design, data collection, or analysis and writing of the manuscript. The authors thank Clare Wilhelm for his assistance with manuscript preparation.

## Author contributions

Conception/design: P.K.P., M.G.K., S.A., J.M. Acquisition, analysis, or interpretation of data: P.K.P., J.L., N.A., R.K., L.A., A.B., C.C., W.M., P.W., D.J.B., W.V.L., J.E.C., M.G.K. All authors have approved the submitted version and have agreed both to be personally accountable for their own contributions and to ensure that questions related to the accuracy or integrity of any part of the work are appropriately investigated, resolved, and the resolution documented in the literature.

## Competing interests

P.K.P. receives compensation for consulting or advisory board participation from Bicara Therapeutics, Inc., EMD Serono, Inc., Takeda Pharmaceuticals, Xencor, Inc, and Mirati; receives honoraria for participation in CME educational programs from PeerVoice, ACE Oncology, Research to Practice, Clinical Care Options, Physicians' Education Resource, Axis Medical Education, and Medscape. J.L. has received honoraria from Targeted Oncology and Physicians' Education Resource. L.A. is a member of the specialty oncology board for ABIM and receives honorarium for this work. P.W. has received consultation fees from Leap Therapeutics. W.V.L. receives institutional research funding from Daiichi Sankyo, Amgen, and Abbvie; has been a compensated consultant for PharmaMar, G1 Therapeutics, AstraZeneca, Jazz Pharmaceuticals. J.E.C. has served as a consultant for Astra Zeneca, Bristol-Myers Squib, Genentech, Merck, Flame Biosciences, Novartis, Regeneron-Sanofi, Guardant Health, and Janssen; has received research funding to her institution from Astra Zeneca, Bristol-Myers Squib, Genentech, and Merck. M.G.K. received personal fees from Novartis, Sanofi, AstraZeneca, Pfizer, Janssen, and Daiichi-Sankyo; received honoraria for participation in educational programs from WebMD, OncLive, Physicians Education Resources, Intellisphere, Continuing Education Alliance, i3 Health, PHSA Corporation, Nexus Health Media, Ideology Health, and AstraZeneca; received travel support from Pfizer and Daiichi-Sankyo; received editorial support from Hoffman La-Roche. M.G.K. is an employee of Memorial Sloan Kettering. Memorial Sloan Kettering has received research funding from The National Cancer Institute (USA), The Lung Cancer Research Foundation and Genentech Roche for research conducted by M.G.K. MSK has licensed testing for EGFR T790M to MolecularMD. The remaining authors declare no competing interests.

## Additional information

[1]Thoracic Oncology Service, Division of Solid Tumor Oncology, Department of Medicine, Memorial Sloan Kettering Cancer Center, New York, NY, USA. [2]Department of Medicine, Weill Cornell Medical College, New York, NY, USA. [3]Division of Biostatistics, College of Public Health, The Ohio State University, Columbus, OH, USA. [4]Institute for Cancer Genetics, Columbia University, New York, NY, USA. [5]Ludwig Center for Cancer Immunotherapy, Memorial Sloan Kettering Cancer Center, New York, NY, USA. [6]Department of Pathology, Memorial Sloan Kettering Cancer Center, New York, NY, USA. [7]Department of Pathology and Cell Biology, Columbia University Irving Medical Center, New York, NY, USA. [8]Herbert Irving Comprehensive Cancer Center, Columbia University, New York, NY, USA. [9]Sloan Kettering Institute, Memorial Sloan Kettering Cancer Center, New York, NY, USA. ✉e-mail: paikp@mskcc.org

