## [Peer Review File · Nature Communications]

Phase I trial of the TNF- α inhibitor certolizumab plus chemotherapy in stage IV lung adenocarcinomasREVIEWER COMMENTS

Reviewer #1 (Remarks to the Author):

Prof. Paik and colleagues presented a phase I trial of TNF- α inhibitor, certolizumab, in combination with chemotherapy as first line treatment for advanced lung adenocarcinoma. The combination strategy was well tolerated. An improved efficacy was observed: with an ORR of 56%, PFS of 7.1 months, OS of 22.6 months. Also, translational investigation confirmed paracrine inflammatory loop found in preclinical studies. Current work provides a cornerstone for further phase II or III validation study, and also for novel combination strategies with anti-PD-1 immunotherapy. However, some issues need to be clarified here.

1. I am considering about the value of this target in the era of chemotherapy. Treatment targeting TNF- α have been widely explored in variety of cancer types. PubMed: 18257397, PubMed: 17674351, doi:10.1016/j.lungcan.2009.06.020, all these studies failed to indicate synergetic effect of TNF- α inhibitor and chemotherapy. In current phase I study, marginal improvement for ORR and PFS was observed in only 18 patients. I don't whether it is worthy for further validation in large sample size. Or we should find new application in the era of immunotherapy.
2. Preclinical work about combination of TNF- α inhibition and cisplatin in animal models is too simple. More investigations about mechanism involved should be included. We need to find out whether changes in humans is in line with those animal models.
3. Fig 1c: 4 groups should be: isotype, isotype+cisplatin, isotype+anti-TNF α , isotype+cisplatin+anti-TNF α
4. More information should be added to figure 2b. We don't know whether all patients have progressed or some patients are still beneficial from treatment.
5. Cytokine assessments were done by comparing with those patients receiving chemotherapy alone. Here, we don't know whether baseline characteristics including driver gene status were balanced between two groups, which might affect cytokine levels. In the control group, patients receive different treatment strategies. 3 received chemotherapy alone and 7 received chemotherapy plus anti-angiogenesis. Different treatments have different impact on cytokine levels. Also the sample size is too small to make it convincing.
6. Annotation for Fig 2B is wrong, the correct is "Duration of Response".
7. Ref.11 is about pancreatic cancer, but not lung cancer.
8. writing mistake in Paragraph 3, page 5: one additional bracket here.

Reviewer #2 (Remarks to the Author):

This paper reported the results from a phase I dose-escalation study with 18 treated patients. They provide descriptive summary statistics, AE table, response and PFS for each patient, and compared some biomarkers (TNF-alpha, S100A8) with a control study. The paper is well written overall, but some issues related to the statistical analyses need to be addressed:

1. My major comment is that Nadir is the lowest value of a biomarker which was measured post-baseline, and it should be considered as a time-dependent covariate in Cox model, and it will be problematic to analyze it as a baseline predictor. Need to clarify which method was used to report HR, 95% and p-value for Nadir TNF-alpha and S100A8 with PFS and OS.
2. Miss some key information
 - a. sample size for the control group was not provide
 - b. methods used to generate the smooth curves on Figure 3 was not provide
 - c. 1 patient without PFS event (should be censored at last day of follow up) but needs to specify this in Figure 2b.
 - d. why "S100A8 levels decreased over time in the treatment group and increased over time in the control group" ? It seems that most of subjects decrease first and reach nadir at some time, then increase or keep. The smooth curve for control might be impacted by some outliers.
 - e. How to calculate mean and AUC for S100A8 ? Using all the available measure to calculate the

mean ? Any time point for AUC (AUC from day 0 to day 28 ?) or all available timepoint ?

f. Minor comment: recommend to add either main title or x-axis label to show left is for treatment group and right is for control group.

Reviewer #3 (Remarks to the Author):

The study would be much stronger if the treatment were evaluated in orthotopic mouse models of human patient lung cancer (International Journal of Cancer 51, 992-995, 1992) rather than the mouse Lewis lung carcinoma.

Reviewer #4 (Remarks to the Author):

In their manuscript, Paul K Paik and co-workers evaluate the impact of combining chemotherapy with anti-TNF in lung adenocarcinomas. The first part of the study suggests the possibility to enhance chemotherapy efficacy with anti-TNF in a mouse lung carcinoma model based on IV administration of Lewis Lung Carcinoma cell line in wild-type mice. The second part depicts clinical data in the context of a phase 1 clinical trial (NCT02120807) in which 18 metastatic lung adenocarcinoma patients have been administered with chemotherapy (cisplatin+pemetrexed) and certolizumab, a humanized Fab' directed against TNF. The clinical data show a good tolerability of the combo and signal of efficacy. The ancillary part is limited and do not provide mechanistic insight. The authors measured plasma TNF and S100A8 concentrations before and after treatment induction, showing a significant decrease of both cytokines along therapy. They evaluated the tumor-infiltrating leukocytes by IHC in tumor samples from patients. No changes were observed upon treatment for all patients but one.

Although the general concept is interesting, the data are too preliminary and do not provide mechanistic insight by which anti-TNF may enhance the chemotherapy efficacy in lung cancer.

1- As depicted in Figure 1, the authors performed only one preclinical experiment with 6-8 mice per group to evaluate the putative benefit of combining chemotherapy and anti-TNF. This experiment is essential for providing the scientific rationale to combine chemotherapy with anti-TNF in patients. This experiment needs to be repeated to further support authors' findings.

2- As depicted in Figure 1, whereas cisplatin alone significantly decreased the metastatic burden, anti-TNF alone did not. The combination of both was associated with a further metastatic burden reduction as compared to cisplatin alone. The results obtained by administering anti-TNF alone are not easy to interpret. Indeed, 4 out of 6 mice exhibited a great reduction of lung metastasis, to the same extent as for mice treated with cisplatin and anti-TNF. How can the authors interpret those findings? Why did 4 out of 6 mice exhibit a dramatic decrease in metastatic burden?

3- Mechanisms involved in the metastatic burden reduction in mice have to be properly investigated under their experimental conditions. What are the molecular and cellular mechanisms involved in the additive effect of chemotherapy and anti-TNF? Is there a direct impact on cancer cells or does it involve alterations of the tumor microenvironment? Since TNF is a major proinflammatory cytokine, which plays pleiotropic functions in the immune system, it would be worthwhile investigating the impact anti-TNF agents (+/- cisplatin) have on the immune responses. Should the authors observe any alteration of the immune responses, they may recapitulate their experiments in relevant immunodeficient mice to clarify the mechanisms by which anti-TNF agents enhance the response to chemotherapy.

4- Concerning the clinical data from 18 lung adenocarcinoma patients, they seem promising with a good tolerability profile. Moreover, 9 patients exhibited objective responses. It remains unclear why the authors have chosen certolizumab as an anti-TNF [and not another anti-TNF drug (ie, infliximab, adalimumab, golimumab, etanercept)]. As stated above, the ancillary part does not allow to propose any molecular and cellular mechanisms to evaluate the impact anti-TNF have in patients (except their impact on TNF and S100A8). Again, deciphering the mechanisms in their preclinical model would allow to generate hypothesis and evaluate their relevance in patient samples. This is essential to better connect both preclinical and clinical data and make this study a strong case.

Minor points

1- In the introduction, the following statement needs to be corrected “Anti-TNF agents have posed no additional safety issue when given to persons with lung cancers, including patients receiving chemotherapy”. Indeed, in reference #11, the authors evaluated the safety and efficacy of infliximab administered with gemcitabine to treat cancer cachexia in 89 patients with stage II-IV pancreatic cancer. No patients affected with lung cancer have been enrolled in this clinical trial.

2- The authors wrote a chapter in the discussion on combining anti-TNF and immune checkpoint blockers in lung cancer patients. This does not make sense since they do not investigate this type of combo in the results section. The results depicted in the manuscript do not provide scientific rationale for combining anti-TNF and immune checkpoint blockers in lung cancer patients.

3- In the discussion, the authors state that anti-TNF agents do not compromise the efficacy of immune checkpoint inhibitors when administered to combat immune-mediated colitis caused by ICIs. Actually, this is still a matter of debate since a study in advanced melanoma patients shows a negative impact of anti-TNF on the OS. “Association of Anti-TNF with Decreased Survival in Steroid Refractory Ipilimumab and Anti-PD1-Treated Patients in the Dutch Melanoma Treatment Registry” (PMID: 31988197).

Reviewer #1 (Remarks to the Author):

Comment #1

Prof. Paik and colleagues presented a phase I trial of TNF- α inhibitor, certolizumab, in combination with chemotherapy as first line treatment for advanced lung adenocarcinoma. The combination strategy was well tolerated. An improved efficacy was observed: with an ORR of 56%, PFS of 7.1 months, OS of 22.6 months. Also, translational investigation confirmed paracrine inflammatory loop found in preclinical studies. Current work provides a cornerstone for further phase II or III validation study, and also for novel combination strategies with anti-PD-1 immunotherapy.

Response #1

Thank you for this summary of our study examining the addition of TNF- α inhibition to combination first line chemotherapy in advanced lung adenocarcinoma. Our positive signal suggests that this strategy should be examined in larger clinical studies.

Comment #2

However, some issues need to be clarified here.

I am considering about the value of this target in the era of chemotherapy. Treatment targeting TNF- α have been widely explored in variety of cancer types. PubMed: 18257397, PubMed: 17674351, doi:10.1016/j.lungcan.2009.06.020, all these studies failed to indicate synergetic effect of TNF- α inhibitor and chemotherapy. In current phase I study, marginal improvement for ORR and PFS was observed in only 18 patients. I don't whether it is worthy for further validation in large sample size. Or we should find new application in the era of immunotherapy.

Response #2

We thank the reviewer for the opportunity to clarify the existing evidence for this study. To clarify, the 3 trials cited are testing TNF- α inhibition in patients with advanced cancers as treatment for cancer cachexia (rather than anti-cancer therapy).

We have summarized the studies as follows:

- 1) The addition of infliximab to gemcitabine for cachexia in patients with pancreatic cancer, a phase 2 study
 - a. N=89, with cachexia as an inclusion criterion
- 2) Etanercept for cancer anorexia and weight loss in individuals with incurable cancers
 - a. N=63, with inclusion criterion of weight loss >2.27kg over 2 months or <20 calories/kg body weight intake
 - b. Outcomes of interest included weight gained, overall survival, and toxicity
- 3) Infliximab for cancer associated weight loss in those who are older/ with poor performance status and non-small cell lung cancer
 - a. N=61, with inclusion criterion of either Eastern Cooperative Oncology Group performance status of 2 or greater or age greater than or equal to 65

We note that while these studies all examined TNF- α inhibition, the inclusion criteria was pan- or other solid tumors and included patients experiencing significant weight loss or were older than the average patient with advanced lung cancer. In addition, efficacy was not the primary objective of these studies. Finally, these studies in cancer cachexia skew towards individuals who have more advanced disease, where an intervention such as TNF- α inhibition for the purposes of improving response to chemotherapy may be too late to show efficacy.

Our phase I study is not only informed by strong preclinical work from Acharyya et al. but also met the prespecified criteria to test for anti-cancer efficacy in a larger patient population. Indeed, while the reviewer notes marginal improvement in ORR and PFS, the study's ORR (56%) and median PFS (7.1 months) and median OS (22.6 months) are in fact substantially longer than the historical efficacy data for cisplatin + pemetrexed alone in patients with newly diagnosed lung adenocarcinoma, where ORR is 30%, median PFS is 4.8-5.3 months, and median OS is 10.3-12.6 months, as noted previously in the Discussion section. We contend that these differences are quite far from marginal.

We fully agree with the reviewer that the next step is to test this in combination with chemo-immunotherapy as that is now becoming the new standard in up-front treatment for patients with NSCLC. Indeed, we briefly mention in the Discussion that the data in this manuscript provided the basis for our new phase 2 study of chemotherapy + nivolumab + certolizumab in early stage NSCLC patients, which is IRB approved and will open to accrual by Q3 2022. We have amended the last paragraph in the Discussion with the new clinicaltrials.gov registry number to reflect this. This is also, in reference to another reviewer's comment, precisely why we included a section in the Discussion about the topic of immunotherapy to contextualize our study.

Comment #3

Preclinical work about combination of TNF- α inhibition and cisplatin in animal models is too simple. More investigations about mechanism involved should be included. We need to find out whether changes in humans is in line with those animal models.

Response #3

The pre-clinical mouse models are, despite the apparent simple photon-flux readout for metastatic burden, quite complicated by nature, requiring immuno-competent NSCLC models that will generate metastatic sites of disease to directly test our hypothesis in an *in vivo* setting that we can manipulate pharmacologically. This rules out the use of typical immortalized human NSCLC cell lines and patient-derived xenografts. The LLC552 cell line we used was specifically generated by the Acharyya lab to exhibit a metastatic phenotype that could be used for experimental testing purposes. Human validation ultimately requires testing our hypothesis in patients, which is why the next immediate step was to execute our phase 1 clinical trial.

However, in response to reviewer comments for additional modeling data, we have added an orthotopic lung cancer model (lung metastatic CMT-167, KRAS mutant background) that demonstrates enhanced cisplatin anti-tumor activity with the addition of anti-TNF- α therapy (new Supplementary Figure 2, added text on page 5 of the Results and page 10 of the Methods section).

In addition, we also provide additional background on the mechanistic data generated by Dr. Acharyya previously. We have generated a new Supplementary Figure 1 that distills the in-depth mechanistic data presented in her 2012 *Cell* paper identifying the elements of the paracrine inflammatory loop and the interplay between cancer cells and the tumor endothelial stroma. Text was also added to the Introduction that delves into this further.

Finally, with regards to validation in humans (our patients), we presented pharmacodynamic data on the impact of certolizumab on TNF- α and S100A8 levels and their association with efficacy outputs (PFS and OS), evidence of promising anti-tumor synergy, and paired patient biopsy data using immunohistochemistry that elucidates changes in immune cell populations. In aggregate, our clinical trial data do suggest that the pre-clinical modeling hypothesis holds and provides enough evidence to move into further phase 2 clinical trial work, as discussed at the end of our manuscript.

Comment #4

Fig 1c: 4 groups should be: isotype, isotype+cisplatin, isotype+anti-TNF α , isotype+cisplatin+anti- TNF α

Response #4

Isotype control labels are listed appropriately in the respective groups in the revised version.

Comment #5

More information should be added to figure 2b. We don't know whether all patients have progressed or some patients are still beneficial from treatment.

Response #5

Thank you for this question regarding the swimmer plot. We have added text in the Figure legend noting that the data reflects progression in all patients save for patient 13, who was censored at the date of last follow-up (also added an asterisk in the Figure noting this).

Comment #6

Cytokine assessments were done by comparing with those patients receiving chemotherapy alone. Here, we don't know whether baseline characteristics including driver gene status were balanced between two groups, which might affect cytokine levels. In the control group, patients receive different treatment strategies. 3 received chemotherapy alone and 7 received chemotherapy plus anti-angiogenesis. Different treatments have different impact on cytokine levels. Also the sample size is too small to make it convincing.

Response #6

Thank you for this comment regarding Figure 3, which graphs TNF- α and S100A8 concentrations. We agree that the absolute concentration of these levels in the blood might be impacted by a variety of baseline and on-treatment features that we are unable to control for. This is an inherent limitation in any clinical study, particularly early phase studies, which is why this promising signal will necessarily be followed by other clinical studies (in particular, our planned phase 2 trial noted in the Discussion). With regard to genotype, we have added a Supplementary File that provides genotype data where available for these patients as reference. The most common alterations in both groups were KRAS mutations in approximately balanced proportion. With regard to angiogenesis inhibition in the control, this was a factor difficult to control for, as at the time bevacizumab was part of the standard first-line regimen at

MSK for patients with lung adenocarcinoma (and so withholding it could not be mandated), though we have no a priori expectations that angiogenesis inhibition would directly affect TNF- α levels (we have added this note to the Discussion section on page 8 as well). Lastly, we note that the particular interest in cytokine levels in our study is not the baseline values, but rather the changes that are associated with chemotherapy and certolizumab. Here, baseline variability because of other unexamined factors is less of a concern, and the magnitude of these changes depicted in Figure 3 are fairly substantial.

Comment #7

Annotation for Fig 2B is wrong, the correct is “Duration of Response”.

Response #7

Thank you for pointing this out. We have corrected what we realize is an incorrect reference to Figure 2B next to the duration of response citation in the text and have moved the reference to Figure 2B in the PFS citation on page 6.

Comment #8

Ref.11 is about pancreatic cancer, but not lung cancer.

Response #8

Thank you for this comment. We have made the following edit to the sentence citing this study:

Edit, Main, Introduction, Page 4:

Anti-TNF- α agents have posed no additional safety issues when given to persons with lung cancers, and in patients with other solid tumors receiving chemotherapy¹¹.

Comment #9

writing mistake in Paragraph 3, page 5: one additional bracket here.

Response #9

We thank the reviewer for noticing this error. We have removed the additional bracket at the end of the sentence.

Reviewer #2 (Remarks to the Author):

Comment #1

This paper reported the results from a phase I dose-escalation study with 18 treated patients. They provide descriptive summary statistics, AE table, response and PFS for each patient, and compared some biomarkers (TNF-alpha, S100A8) with a control study. The paper is well written overall, but some issues related to the statistical analyses need to be addressed:

My major comment is that Nadir is the lowest value of a biomarker which was measured post-baseline, and it should be considered as a time-dependent covariate in Cox model, and it will be problematic to analyze it as a baseline predictor. Need to clarify which method was used to report HR, 95% and p-value for Nadir TNF-alpha and S100A8 with PFS and OS.

Response #1

We have amended the manuscript to include the PFS and OS regression for TNF-alpha and S100A8 as time-dependent covariates, in addition to the time-independent model. These analyses are being done for the purposes of broad biomarker exploration, and so we agree that the inclusion of a time-dependent covariate as part of this makes sense to round out the analysis (pages 6 and 13). We are retaining the time-independent model for the purposes of hypothesis generation, as it is not clear that there will be a clear and consistent temporal relationship between changes in these biomarkers and the effect of certolizumab, given the dynamic and variable nature of the effects of the overall anti-cancer regimen involved and disease status of the patients.

Comment #2

Miss some key information

1. sample size for the control group was not provide
 - a. This is provided in the Methods section (N=10)
2. methods used to generate the smooth curves on Figure 3 was not provide
 - a. The smooth curves are cubic splines with 4 knots. We have added this to the Methods section on page 13
3. 1 patient without PFS event (should be censored at last day of follow up) but needs to specify this in Figure 2b.
 - a. We have added notation that the data are censored to the last day of follow-up in Figure 2B for that 1 patient.
4. why “S100A8 levels decreased over time in the treatment group and increased over time in the control group” ? It seems that most of subjects decrease first and reach nadir at some time, then increase or keep. The smooth curve for control might be impacted by some outliers.
 - a. This is indeed what we would have anticipated as the dynamic in S100A8 between the experimental and control groups. The increase in the control group is hypothesized to occur, as we would expect that without TNF- α suppression, S100A8 levels would go up (which we in fact see), as the distal cytokine in the paracrine inflammatory loop. Conversely, with TNF- α suppression of the inflammatory loop in the experimental group, S100A8 would go down (which is also what is seen).
 - b. The smooth curve was utilized to better see trends in the data, taking into consideration the variability that might be created by outliers.
5. How to calculate mean and AUC for S100A8 ? Using all the available measure to calculate the mean ? Any time point for AUC (AUC from day 0 to day 28 ?) or all available timepoint ?
 - a. We utilized all available timepoints. The AUC is standardized by dividing it by the length of trajectory. These details have been added to the Statistical Analysis section of the manuscript. (page 13)
6. Minor comment: recommend to add either main title or x-axis label to show left is for treatment group and right is for control group.
 - a. Titles have been to figure 3 to clarify which graphs show patients in the treatment group vs. those in the control group.

Reviewer #3 (Remarks to the Author):

Comment #1

The study would be much stronger if the treatment were evaluated in orthotopic mouse models of human patient lung cancer (International Journal of Cancer 51, 992-995, 1992) rather than the mouse Lewis lung carcinoma.

Response #1

Because patient-derived xenografts require immunodeficient mice, this model system could not adequately recapitulate the conditions required to investigate the proposed mechanism of action which requires immunocompetent models which also generate metastatic sites of disease. Similarly, traditional human cell line xenografts require immunodeficient mice, limiting the ability to model our specific proposed mechanism.

Hence our selection of an LLC mouse model, which can grow in immunocompetent C57/B16 mice, and which generates widespread metastatic sites of disease through intracardiac injection and dissemination. We have, in response to the reviewer comment expanded our model systems, tested a new orthotopically implanted lung-metastatic CMT-167 *KRAS* mutant lung cancer model, which also grows in immunocompetent mice, under the same experimental conditions. This also demonstrates efficacy of anti-TNF- α therapy added to cisplatin chemotherapy (new Supplementary Figure 2).

Reviewer #4 (Remarks to the Author):

Comment #1

In their manuscript, Paul K Paik and co-workers evaluate the impact of combining chemotherapy with anti-TNF in lung adenocarcinomas. The first part of the study suggests the possibility to enhance chemotherapy efficacy with anti-TNF in a mouse lung carcinoma model based on IV administration of Lewis Lung Carcinoma cell line in wild-type mice. The second part depicts clinical data in the context of a phase 1 clinical trial (NCT02120807) in which 18 metastatic lung adenocarcinoma patients have been administered with chemotherapy (cisplatin+pemetrexed) and certolizumab, a humanized Fab' directed against TNF. The clinical data show a good tolerability of the combo and signal of efficacy. The ancillary part is limited and do not provide mechanistic insight. The authors measured plasma TNF and S100A8 concentrations before and after treatment induction, showing a significant decrease of both cytokines along therapy. They evaluated the tumor-infiltrating leukocytes by IHC in tumor samples from patients. No changes were observed upon treatment for all patients but one.

Although the general concept is interesting, the data are too preliminary and do not provide mechanistic insight by which anti-TNF may enhance the chemotherapy efficacy in lung cancer.

Response #1

We thank the reviewer for this summary of our study and the interest in our clinical trial.

The mechanistic insight into the paracrine inflammatory loop at the heart of this concept was previously generated by Dr. Acharyya in *Cell*. This paper provides in-depth experimental data that identify the key elements of the paracrine inflammatory loop, including the cytokines involved and the interaction between the tumor and normal stroma. Recapitulating these data was thus not the priority of our phase 1 study; rather the focus was on clinically translating her work by taking the first required step, which is a phase 1 trial to demonstrate safety and to generate preliminary pharmacodynamic and efficacy data in support of the hypothesized mechanism of action. The study was successful in both endeavors. The reviewer is absolutely correct- phase 1 data of this kind cannot stand alone in making definitive declarations about either pharmacodynamics or efficacy, and so the study requires further validation in phase 2 trials.

We note this in the final paragraph of the Discussion, and that the next step clinical step that is building on the results of the phase 1 detailed in this manuscript, which is a phase 2 trial of certolizumab + chemoimmunotherapy in early stage NSCLC patients, which will provide a much greater material and sample size capacity (resected primary lung cancer tumors rather than small needle biopsies) to generate hypothesis validating data.

Comment #2

As depicted in Figure 1, the authors performed only one preclinical experiment with 6-8 mice per group to evaluate the putative benefit of combining chemotherapy and anti-TNF. This experiment is essential for providing the scientific rationale to combine chemotherapy with anti-TNF in patients. This experiment needs to be repeated to further support authors' findings.

Response #2

We have repeated the LLC experiment in Figure 1 with 7 additional mice in each condition as requested with similar results (new Figure 1C) and in addition, in response to the reviewer's comment, added another model utilizing an orthotopic lung-metastatic CMT-167 *KRAS* mutant lung cancer line that additionally validates these findings (new Supplementary Figure 2).

Comment #3

As depicted in Figure 1, whereas cisplatin alone significantly decreased the metastatic burden, anti-TNF alone did not. The combination of both was associated with a further metastatic burden reduction as compared to cisplatin alone. The results obtained by administering anti-TNF alone are not easy to interpret. Indeed, 4 out of 6 mice exhibited a great reduction of lung metastasis, to the same extent as for mice treated with cisplatin and anti-TNF. How can the authors interpret those findings? Why did 4 out of 6 mice exhibit a dramatic decrease in metastatic burden?

Response #3

We thank the reviewer for noticing this. We repeated the experiment shown in Figure 1 with 7 additional mice in each experimental condition, summarized in a revised Figure 1. This demonstrates a similar pattern of response overall, with remarkable synergy between cisplatin + anti-TNF therapy to blunt metastatic propagation, and also shows that the effect of anti-TNF therapy alone is much more consistent and modest across all 7 mice. It is not clear what led some of the mice in the first experiment to have a discordant result with TNF- α treatment alone. One implication and explanation of the variability seen in the first experimental run could be differing levels of endogenous TNF- α present in disparate mice, and thus differing levels of both autocrine signaling and paracrine signaling that might be present that cannot be controlled for between animals. Prior metastatic colonization modeling work has suggested that TNF- α plays a more proximal role in the metastatic process in other contexts, and some genetically engineered mouse models of lung cancer are marked by highly inflamed tumors, exhibiting upregulation of TNF- α which is presumably involved in the carcinogenic process. We have added mention of this in the Discussion section on page 8.

Comment #4

Mechanisms involved in the metastatic burden reduction in mice have to be properly investigated under their experimental conditions. What are the molecular and cellular mechanisms involved in the additive effect of chemotherapy and anti-TNF? Is there a direct impact on cancer cells or does it involve alterations of the tumor microenvironment? Since TNF is a major proinflammatory cytokine, which plays pleiotropic functions in the immune system, it would be worthwhile investigating the impact anti-TNF agents (+/- cisplatin) have on the immune responses. Should the authors observe any alteration of the immune responses, they may recapitulate their experiments in relevant immunodeficient mice to clarify the mechanisms by which anti-TNF agents enhance the response to chemotherapy.

Response #4

The molecular and cellular mechanisms involved in the additive effect of chemotherapy and anti-TNF were explored in detail in Dr. Acharyya's 2012 *Cell* publication describing the paracrine inflammatory loop that grounds the phase 1 trial. The purpose of our work was to begin to clinically translate her detailed work into a phase 1 trial, to demonstrate the safety of adding certolizumab to chemotherapy and

to show, in a preliminary fashion, an effect on relevant pharmacodynamics and efficacy, which the trial has done.

It was beyond the scope of this work and would have been redundant to repeat the extensive experimental work performed by Dr. Acharyya in her key paper. This said, we understand that while we as authors are quite familiar with the details of this work, our manuscript does not sufficiently describe this for the new reader. We have amended the Introduction with more details on the molecular and cellular elements of the paracrine inflammatory loop and provide a new Supplementary Figure 1 which demonstrates these elements for the reader.

Comment #5

Concerning the clinical data from 18 lung adenocarcinoma patients, they seem promising with a good tolerability profile. Moreover, 9 patients exhibited objective responses. It remains unclear why the authors have chosen certolizumab as an anti-TNF [and not another anti-TNF drug (ie, infliximab, adalimumab, golimumab, etanercept)]. As stated above, the ancillary part does not allow to propose any molecular and cellular mechanisms to evaluate the impact anti-TNF have in patients (except their impact on TNF and S100A8). Again, deciphering the mechanisms in their preclinical model would allow to generate hypothesis and evaluate their relevance in patient samples. This is essential to better connect both preclinical and clinical data and make this study a strong case.

Response #5

Thank you for this question regarding why certolizumab was the chosen TNF- α inhibitor in the clinical study. Certolizumab was the ideal candidate for this for this study because it is the only TNF- α antagonist that is built on an Fab fragment of humanized TNF- α antibody that lacks the Fc region. This avoids both complement-dependent cytotoxicity and antibody dependent cell mediated cytotoxicity as confounders which are seen with all other anti-TNF agents. We have added note of this in in the Introduction on page 4. From a modeling perspective, certolizumab could not be utilized as it is ineffective in blocking murine TNF- α . This points to the limitation of mouse modeling of the human experience.

Comment #6

Minor points

In the introduction, the following statement needs to be corrected “Anti-TNF agents have posed no additional safety issue when given to persons with lung cancers, including patients receiving chemotherapy”. Indeed, in reference #11, the authors evaluated the safety and efficacy of infliximab administered with gemcitabine to treat cancer cachexia in 89 patients with stage II-IV pancreatic cancer. No patients affected with lung cancer have been enrolled in this clinical trial.

Response #6

Thank you for this comment. We have made edits in response to Reviewer 1 Comment 8 that remedy this.

Comment #7

The authors wrote a chapter in the discussion on combining anti-TNF and immune checkpoint blockers in lung cancer patients. This does not make sense since they do not investigate this type of combo in the results section. The results depicted in the manuscript do not provide scientific rationale for combining anti-TNF and immune checkpoint blockers in lung cancer patients.

Response #7

We thank the reviewer for asking about combining TNF- α inhibitors with immune checkpoint inhibitors. We agree that our study does not directly address the addition of immune checkpoint inhibitors into the tested regimen. However, it is important to note that the standard of care for lung cancer has shifted rapidly over the past several years and immune checkpoint inhibitors are now standard treatment in the

first line setting for many patients with NSCLC. This is something that Reviewer 1 also points out. Our study preceded the approval of these treatments and occurred when chemotherapy alone was the standard of care. We thus found it important to contextualize the current study by adding this section into the Discussion, which also fulfills Reviewer 1's commentary on the topic. To be sure, the purpose of this discussion is to frame what our next clinical step is, understanding that as the Reviewers have pointed out, our phase 1 data are preliminary and require further clinical validation. The next step is to test certolizumab coupled to chemo-immunotherapy in a larger phase 2 trial in early stage NSCLC patients, which will allow us to directly test the effect of TNF- α blockade on metastatic recurrence and to generate much more correlative data given the larger sample size of this study and availability of whole primary tumor resection specimens to test, rather than small needle biopsies.

Comment #8

In the discussion, the authors state that anti-TNF agents do not compromise the efficacy of immune checkpoint inhibitors when administered to combat immune-mediated colitis caused by ICIs. Actually, this is still a matter of debate since a study in advanced melanoma patients shows a negative impact of anti-TNF on the OS. "Association of Anti-TNF with Decreased Survival in Steroid Refractory Ipilimumab and Anti-PD1-Treated Patients in the Dutch Melanoma Treatment Registry" (PMID: 31988197).

Response #8

Thank you for bringing up this study. This is a difficult question to answer clinically because of survivor bias. Individuals who develop immune-mediated colitis have had to be on these treatments long enough to be at risk for colitis. Therefore, a well-conducted, robust retrospective study examining irAEs should provide multiple analyses that consistently show different adjustments to account for survivor bias (from conservative to more liberal adjustments) lead to similar findings. Patients with immune-mediated colitis on TNF- α inhibitors tend to be a sicker patient population than those not on these treatments and may require hospitalization for hydration and electrolyte disturbances given the quantity and frequency of diarrhea. Steroid use is also not benign, and these patients often need to be on high doses of steroids for long tapers. In the specific study mentioned, which uses registry-level data, the investigators did not have enough granularity to disentangle the contribution of the side effects from high dose prolonged steroid use and a sicker patient population, from the actual harm of TNF- α inhibitors.

REVIEWER COMMENTS

Reviewer #1 (Remarks to the Author):

The authors gave a good explanation to all questions raised by reviewers point by point. The research is interesting, but clinical significance is limited in the era of immunotherapy.

Reviewer #2 (Remarks to the Author):

thanks for the thorough reply. The authors addressed my questions appropriately. Looks good.

Reviewer #4 (Remarks to the Author):

The first version of the paper has been amended with an additional preclinical lung cancer model orthotopically grafted to immunocompetent mice (i.e., CMT-167 cells). The results show convincing evidence that anti-TNF enhances the chemotherapy efficacy in this model. Moreover, the authors repeated the experiments using the LLC carcinoma model in immunocompetent mice. The latter experiment provides more convincing results than the experiment depicted in the first version of the manuscript. However, the molecular and cellular mechanisms remain to be determined under their experimental conditions. With no clear mechanisms, the study lacks of novelty, despite the clinical trial. Moreover, the authors claim they already established the mechanisms in breast cancer models [ref #5 of the manuscript: Acharyya S, et al. A CXCL1 paracrine network links cancer chemoresistance and metastasis. *Cell* 150, 165-178 (2012).]. One should note that the impact of combining anti-TNF (ie, infliximab) to chemotherapy has been evaluated using human MDA231-LM2 orthotopically grafted in the mammary fat pad of immunodeficient NOD-SCID mice (Fig. 6 of ref #5). Mechanisms observed upon grafting human breast cancer cell lines in immunodeficient mice cannot be extrapolated in immunocompetent mouse lung carcinoma models. This comment is all the more relevant since anti-TNF modulate both innate and adaptive immune responses. TNF signaling likely modulates the differentiation status of cancer cells, leading to EMT process and expression of anti-apoptotic proteins for instance. Depicting the graphical abstract from a paper published in 2012 is not enough to decipher the mechanisms by which anti-TNF enhance the chemotherapy efficacy in immunocompetent mouse lung carcinoma models.

Major comments:

1- Mechanisms involved in the metastatic burden reduction in mice have to be properly investigated under their experimental conditions. What are the molecular and cellular mechanisms involved in the additive effect of chemotherapy and anti-TNF? Is there a direct impact on cancer cells or does it involve alterations of the tumor microenvironment? Since TNF is a major proinflammatory cytokine, which plays pleiotropic functions in the immune system, it would be worthwhile investigating the impact anti-TNF agents (+/- cisplatin) have on the immune responses. Should the authors observe any alteration of the immune responses, they may recapitulate their experiments in relevant immunodeficient mice to clarify the mechanisms by which anti-TNF agents enhance the response to chemotherapy.

2- Except the data on TNF and S100A8 in plasma samples, the ancillary part of the clinical trial does not bring any insights into the mechanisms associated with the combo of chemotherapy and certolizumab. The IHC experiments in tumor samples from patients do not allow to draw any conclusion. Some IHC data are surprising. For instance, the authors comment on the FOXP3+ cells increase upon treatment induction in patient 3, whose tumor shows a strong increase in tumor-infiltrating leukocytes. What are those FOXP3+ cells since there is no CD4+ cells before and after treatment induction? Why the authors do not comment on the 6-to-7-fold increase of CD8+ cells? Why the authors did not monitor the S100A8 levels on tumor samples?

Minor comments:

1- Concerning the description of Fig. 1C, "Treatment with cisplatin, and TNF- α antibody to a lesser

extent, reduced metastasis over isotype-control treated mice ($p=0.0007$; Fig. 1C), with the combination of cisplatin plus anti-TNF- α antibody further lowering metastatic burden ($p= 0.03$)” The p values depicted in the text are not consistent with the ones from the figure legend, “Values are mean \pm standard error. * $p = 0.0379$, ** $p =0,0012$, ns = not significant”. Moreover, there are 4 ** in the Figure. Please homogenize and clarify.

2- Concerning the association between PFS and TNF- α and S100A8 as time dependent covariates. The authors state “Time-dependent modeling was also performed to explore associations between TNF- α and S100A8 and PFS and OS, which showed trends towards worse PFS with increasing TNF- α and worse OS with increasing TNF- α and S100A8 (Supplementary Table 2).” According to supp table 2, the only variable that seems to be almost significant is the variation of TNF concentration and OS (HR 1.58 $p=0.069$). The other associations are far from being significant.

Reviewer #1 (Remarks to the Author):**Comment #1**

The authors gave a good explanation to all questions raised by reviewers point by point. The research is interesting, but clinical significance is limited in the era of immunotherapy.

Response #1

We thank the reviewer for their feedback and note that the clinical significance in the era of immunotherapy will be addressed by our IRB-approved, soon to be open to accrual phase 2 trial of platinum doublet chemotherapy + nivolumab + certolizumab in patients with early-stage non-small cell lung cancers as mentioned in the Discussion section. The current manuscript was the necessary first-step in mapping out our overall clinical strategy, by demonstrating safety, preliminary efficacy, and pharmacodynamic effects with the addition of certolizumab to platinum doublet chemotherapy in advanced NSCLC patients prior to moving our work into the curative, early stage setting. We anticipate sharing exciting results from this study in the next 1-2 years through conferences, culminating in a future publication.

Reviewer #2 (Remarks to the Author):**Comment #1**

Thanks for the thorough reply. The authors addressed my questions appropriately. Looks good.

Response #1

We thank the reviewer for their time spent in consideration of our manuscript.

Reviewer #4 (Remarks to the Author):**Comment #1-2**

The first version of the paper has been amended with an additional preclinical lung cancer model orthotopically grafted to immunocompetent mice (i.e., CMT-167 cells). The results show convincing evidence that anti-TNF enhances the chemotherapy efficacy in this model. Moreover, the authors repeated the experiments using the LLC carcinoma model in immunocompetent mice. The latter experiment provides more convincing results than the experiment depicted in the first version of the manuscript. However, the molecular and cellular mechanisms remain to be determined under their experimental conditions. With no clear mechanisms, the study lacks of novelty, despite the clinical trial. Moreover, the authors claim they already established the mechanisms in breast cancer models [ref #5 of the manuscript: Acharyya S, et al. A CXCL1 paracrine network links cancer chemoresistance and metastasis. *Cell* 150, 165-178 (2012).]. One should note that the impact of combining anti-TNF (ie, infliximab) to chemotherapy has been evaluated using human MDA231-LM2 orthotopically grafted in the mammary fat pad of immunodeficient NOD-SCID mice (Fig. 6 of ref #5). Mechanisms observed upon grafting human breast cancer cell lines in immunodeficient mice cannot be extrapolated in immunocompetent mouse lung carcinoma models. This comment is all the more relevant since anti-TNF modulate both innate and adaptive immune responses. TNF signaling likely modulates the differentiation status of cancer cells, leading to EMT process and expression of anti-apoptotic proteins for instance. Depicting the graphical abstract from a paper published in 2012 is not enough to decipher the mechanisms by which anti-TNF enhance the chemotherapy efficacy in immunocompetent mouse lung carcinoma models.

Mechanisms involved in the metastatic burden reduction in mice have to be properly investigated under their experimental conditions. What are the molecular and cellular mechanisms involved in the additive effect of chemotherapy and anti-TNF? Is there a direct impact on cancer cells or does it involve alterations of the tumor microenvironment? Since TNF is a major proinflammatory cytokine, which plays

pleiotropic functions in the immune system, it would be worthwhile investigating the impact anti-TNF agents (+/- cisplatin) have on the immune responses. Should the authors observe any alteration of the immune responses, they may recapitulate their experiments in relevant immunodeficient mice to clarify the mechanisms by which anti-TNF agents enhance the response to chemotherapy.

Response #1-2

We appreciate Reviewer 4's thorough review of the manuscript. We agree that studies in immunodeficient mice cannot be extended to immunocompetent models. However, we note that studies of both immunodeficient *and* immunocompetent mice were indeed included in Dr. Acharyya's manuscript (Acharyya S, et al., *Cell* 150, 165-178 [2012]). Results from her study show that knockdown of CXCL1/2 in immunodeficient (Fig. 1 panels G-J) and immunocompetent (Fig. 1 panels C-F) breast cancer models similarly blunts the growth of breast cancer metastases to the lung. This work establishes that the CXCL1 paracrine network promotes lung metastasis in both immunodeficient and immunocompetent models, and that there has not been a mechanistic extrapolation on our part in either setting as a result.

We thus respectfully contend that this major comment is an inaccurate characterization of our work, and that the subsequently requested near-total recapitulation of our co-author Dr. Acharyya's prior study is outside of the scope of what would be reasonably considered. To wit, the initial mechanistic framework was well-established in Dr. Acharyya's *Cell* manuscript, and this observation was extended by us into a lung cancer framework for a purpose: to facilitate translation of this mechanism of action into patient care. This is an important distinction, and the *raison d'être* of our study. The next logical step forward was not to reproduce Dr. Acharyya's original *Cell* work in its "molecular and cellular" mechanistic totality in different model systems. Such work would take years, and would ultimately be derivative and static (indeed, from this standpoint the pre-clinical studies in our manuscript are the least novel part of our paper; it is the clinical trial as fruit born of our pre-clinical validation in lung cancer that is novel).

As such, our approach was to ensure, as efficiently as possible, that the paracrine inflammatory loop could be therapeutically targeted in lung cancer models through a clinically available approach, and then to translate this into a clinical trial as quickly as possible. It bears stating that from an overarching perspective, pre-clinical studies in cancer are never ends to themselves- they are always done in service to and with the hope that they will lead to therapeutic next-steps in patient care. Thus, we feel that Reviewer 4's perspective is too narrow, and unreasonably downplays the importance and centrality of the phase 1 clinical trial reported in our study. Indeed, the efficacy and correlative data reported from our clinical trial validate the pre-clinical results and is precisely why we are moving forward with another IRB-approved clinical trial in early stage disease that builds on the current work submitted to *Nature Communications*. We also note that in our revision response, all of Reviewer 4's prior points were addressed, and that the not insignificant time spent in revision was to generate a whole new lung cancer model to further validate our approach in response to the prior review.

In response to Reviewer 4's major comments, we have expanded the Discussion to further clarify that our study integrates into an existing data-driven framework, and to more explicitly explain the intentional pre-clinical experimental tact taken in order to facilitate the execution of our phase 1 clinical trial (see first paragraph in the Discussion).

"We note that the overarching objective of our study was to efficiently demonstrate that this paracrine inflammatory loop, which we had previously characterized in exacting mechanistic detail, could be therapeutically targeted using a clinically available drug in 1) lung cancer models in order to reduce metastatic burden, and 2) patients with lung cancer in order to improve clinical outcomes."

Comment #3

Except the data on TNF and S100A8 in plasma samples, the ancillary part of the clinical trial does not bring any insights into the mechanisms associated with the combo of chemotherapy and certolizumab. The IHC experiments in tumor samples from patients do not allow to draw any conclusion. Some IHC data are surprising. For instance, the authors comment on the FOXP3+ cells increase upon treatment induction in patient 3, whose tumor shows a strong increase in tumor-infiltrating leukocytes. What are those FOXP3+ cells since there is no CD4+ cells before and after treatment induction? Why the authors do not comment on the 6-to-7-fold increase of CD8+ cells? Why the authors did not monitor the S100A8 levels on tumor samples?

Response #3

On reassessment of the IHC scoring, the CD4 IHC staining failed on the sample run for patient 3. This was transferred to the Supplementary Table as a “0” score, but should have been marked as N/A. This has been amended in Supplementary Table 3. We understand how this has caused confusion but note that FOXP3 is a more specific marker for Tregs than CD4, and so does not require modification of the Discussion text on this topic. Regarding the IHC results, while the data are interesting and provide a unique window into the immune microenvironment, because the sample sizes are relatively small, we have been careful to restrict commentary on the results to avoid overinterpretation of the data. It is unclear what the increase in CD8+ T-cells reflects, and so no productive commentary on this is provided in the Discussion. Regarding S100A8 IHC, we do not have a validated S100A8 antibody for use in our Pathology Core laboratory for human tissue, and so could not probe for this.

Comment #4

Concerning the description of Fig. 1C, “Treatment with cisplatin, and TNF- α antibody to a lesser extent, reduced metastasis over isotype-control treated mice ($p=0.0007$; Fig. 1C), with the combination of cisplatin plus anti-TNF- α antibody further lowering metastatic burden ($p=0.03$)” The p values depicted in the text are not consistent with the ones from the figure legend, “Values are mean \pm standard error. * $p = 0.0379$, ** $p = 0.0012$, ns = not significant”. Moreover, there are 4 ** in the Figure. Please homogenize and clarify.

Response #4

We thank the reviewer for pointing this discrepancy out and have aligned the notations in the text and figure.

Comment #5

Concerning the association between PFS and TNF- α and S100A8 as time dependent covariates. The authors state “Time-dependent modeling was also performed to explore associations between TNF- α and S100A8 and PFS and OS, which showed trends towards worse PFS with increasing TNF- α and worse OS with increasing TNF- α and S100A8 (Supplementary Table 2).” According to supp table 2, the only variable that seems to be almost significant is the variation of TNF concentration and OS (HR 1.58 $p=0.069$). The other associations are far from being significant.

Response #5

The common term used to describe directionality of a test that does not meet statistical significance is “trend towards”. This is the phrase that we used to describe the time-dependent modeling for TNF- α and S100A8 for PFS and OS as shown in Supplementary Table 2. We are aware that the tests did not meet statistical significance, which is why the precise language used in the text was chosen (again, “trend towards”). We contend that no modification of the text is thus required.